# Comparison of porcine corneal decellularization methods and importance of preserving corneal limbus through decellularization

Abdulkadir Isidan[1], Shaohui Liu[2], Angela M. Chen[1], Wenjun Zhang[1], Ping Li[1], Lester J. Smith[3,4], Hidetaka Hara[5], David K. C. Cooper[5], Burcin Ekser[1]*

1 Transplant Division, Department of Surgery, Indiana University School of Medicine, Indianapolis, Indianapolis, United States of America, 2 Department of Ophthalmology, Glick Eye Institute, Indiana University School of Medicine, Indianapolis, Indianapolis, United States of America, 3 3D Bioprinting Core, Indiana University School of Medicine, Indianapolis, Indianapolis, United States of America, 4 Department of Radiology and Imaging Sciences, Indiana University of School of Medicine, Indianapolis, Indianapolis, United States of America, 5 Xenotransplantation Program, Department of Surgery, University of Birmingham at Alabama, Birmingham, Alabama, United States of America

* bekser@iupui.edu

**Data Availability Statement:** All relevant data are within the paper and its Supporting information files.

## Abstract

### Background

The aim of this study is to compare the three previously applied, conventional porcine corneal decellularization methods and to demonstrate the importance of preserving the corneal limbus through decellularization.

### Methods

Fresh, wild-type (with or without) limbus porcine corneas were decellularized using three different methods, including (i) sodium dodecyl sulfate (SDS), (ii) hypertonic saline (HS), and (iii) $N_2$ gas (NG). Post-treatment evaluation was carried out using histological, residual nuclear material, and ultrastructural analyses. Glycerol was used to help reduce the adverse effects of decellularization. The corneas were preserved for two weeks in cornea storage medium.

### Results

All three decellularization methods reduced the number of keratocytes at different rates in the stromal tissue. However, all methods, except SDS, resulted in the retention of large numbers of cells and cell fragments. The SDS method (0.1% SDS, 48h) resulted in almost 100% decellularization in corneas without limbus. Low decellularization capacity of the NG method (<50%) could make it unfavorable. Although HS method had a more balanced damage-decellularization ratio, its decellularization capacity was lower than SDS method. Preservation of the corneoscleral limbus could partially prevent structural damage and edema, but it would reduce the decellularization capacity.

**Funding:** Work on xenotransplantation in the Xenotransplantation Research Laboratory at Indiana University has been supported by internal funds of the Department of Surgery, in part, with support by the Board of Directors of the Indiana University Health Values Fund for Research Award (VFR-457-Ekser), the Indiana Clinical and Translational Sciences Institute, funded in part by Grant #UL1TR001108 from the National Institutes of Health, National Center for Advancing Translational Sciences, Clinical and Translational Sciences Award. The funders had no role in study design, data collection and analysis, decision to publish, or preparation of the manuscript.

**Competing interests:** The authors have declared that no competing interests exist.

**Abbreviations:** DPBS, Dulbecco's phosphate-buffered saline; ECM, Extracellular matrix; H&E, Hematoxylin-eosin; HS, Hypertonic saline; NG, N2 gas; NPCs, Native porcine corneas; SD, Standard deviation; SDS, Sodium dodecyl sulfate; TEM, Transmission electron microscopy; TX, Triton X-100; WT, Wild-type.

## Conclusion

Our results suggest that SDS is a very powerful decellularization method, but it damages the cornea irreversibly. Preserving the corneoscleral limbus reduces the efficiency of decellularization, but also reduces the damage.

## 1. Introduction

It is estimated that there are 36 million people worldwide who are blind [1]. Of these patients, corneal blindness accounts for 5–12% of all the cases [2–4]. Furthermore, due to corneal trauma and ulcers, 1.5–2.0 million new individuals lose sight in one eye every year [5]. Currently, corneal allo-transplantation is the only approved curative option when the cornea becomes opaque [6]. Unfortunately, there is a large discrepancy between the supply and demand of corneal tissue in many countries. In a broad perspective, there are four ways that could alleviate the cornea shortage worldwide: 1) prevention; 2) development of strategies for corneal healing; 3) enhancing the organ donation numbers; and 4) development of tissue grafts that mimic the cornea. Although a decreasing trend was observed in the last decades with the action plans of the World Health Organization, preventable causes of corneal blindness (e.g. trachoma, onchocerciasis, keratomalacia) still account for approximately 80% of all corneal blindness [1, 4, 7].

Currently, deceased human donors are the only source for corneal grafts, but the numbers are far away from supplying the demand. Recently, tissue engineering strategies have emerged that could solve the corneal shortage. 3D-bioprinting and xenotransplantation (e.g., pig cornea-to-human) are two major strategies that have potential to solve the corneal shortage. While the heterogeneic architecture of the cornea could be a problem for 3D-bioprinting, conservation of the extracellular matrix (ECM) proteins and mechanical properties are a most significant advantage for corneal xenotransplantation [8]. The availability and ability of genetic engineering of the donor pig to address the immunological target make xenotransplantation more attractive [9]. However, immune rejection still remains to be overcome despite the use of genetically-engineered pig corneas, cells, and/or biomaterials. Although decellularization techniques aim to avoid this immune reaction, the problem with a decellularization strategy is that almost all techniques have variable adverse effects on the corneal tissue.

In the past, many different decellularization protocols for porcine corneal tissues have been reported using biological, physical, and chemical compounds [10–12]. Besides conventional methods there are newer methods such as high hydrostatic pressure, supercritical $CO_2$, phospholipase $A_2$, formic acid etc. and newer agents that used to improve conventional methods such as benzonase, sodium N-lauroyl glutamate, supernuclease etc. [13–18]. An ideal corneal decellularization method should have high cell removal and antigenicity elimination capacities, while maintaining the native histo-architecture and ECM. In the present study, we evaluated and compared side-by-side three conventional decellularization protocols of full thickness pig corneas with or without the limbus in order to assess decellularization efficiency and undesirable effects of each decellularization protocol as well as the importance of preserving the corneal limbus.

## 2. Materials and methods

### 2.1. Decellularization protocols

Fresh, wild-type (WT) porcine eyeballs were purchased from a local slaughterhouse. Eyeballs were enucleated within 1 hour after death and transported with 1% Antibiotic-Antimycotic in

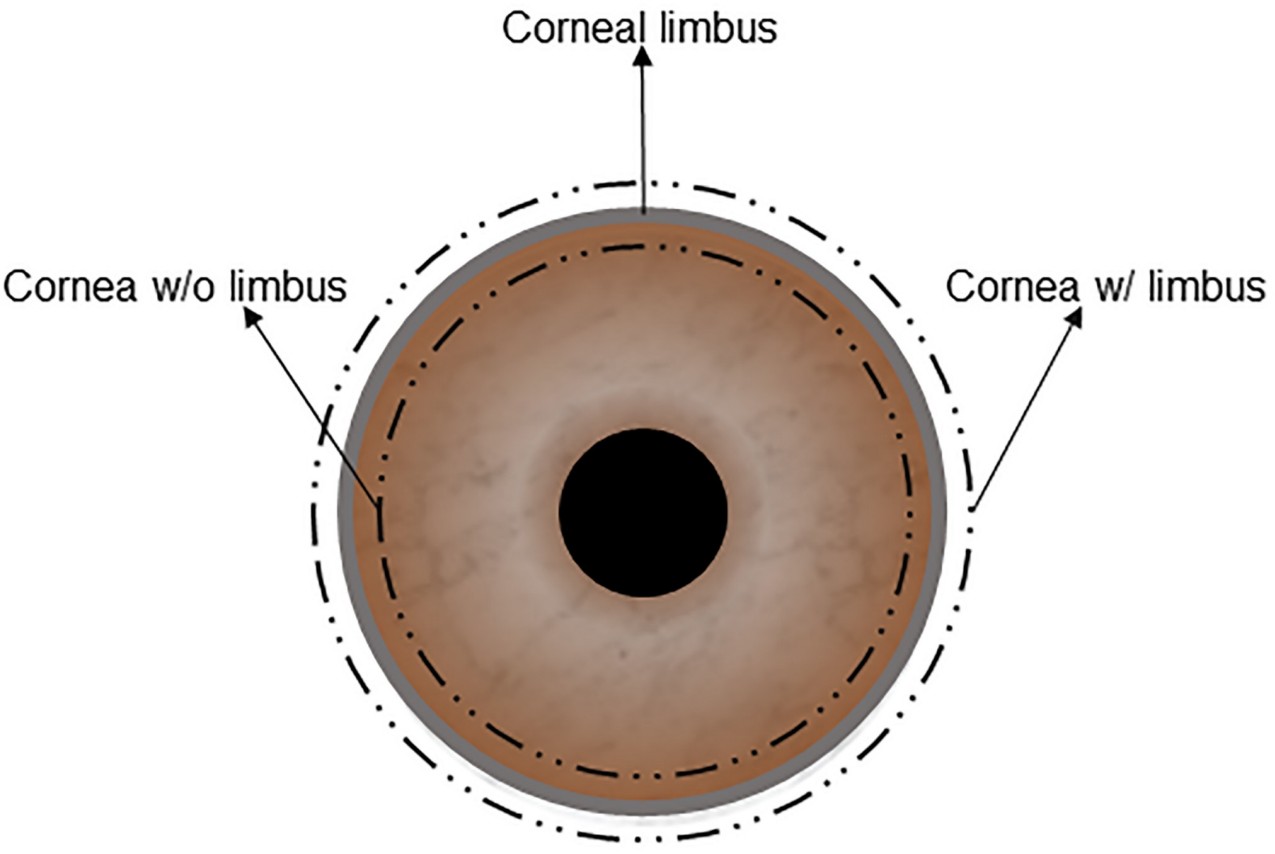

**Fig 1. A figuration of a corneal graft with limbus or without the limbus.**

Dulbecco's phosphate-buffered saline (DPBS). The corneas were excised either with at least 1 mm of scleral tissue (cornea w/ limbus) or at least 1 mm within the corneal limbus (cornea w/o limbus) (Fig 1). All corneas were then rinsed with 10% Antibiotic-Antimycotic solution for 15 minutes. Full-thickness WT native porcine corneas (NPCs) were used in these experiments. NPCs were divided into four groups (n = 6 in each group; 3 with limbus and 3 without limbus), and decellularization protocols were performed as follows: 1) Untreated NPCs (control group). 2) SDS (Sodium Dodecyl Sulfate) Method: corneas were immersed in 0.1% SDS (GibcoBrl, Grand Island, NY) for 48 hours at room temperature with 300 rpm of continuous shaking and decellularization solution renewed every 12 hours [19]. 3) HS (Hypertonic Saline) Method: corneas were immersed in ultrapure water for 12 hours and subsequently treated with 2 M NaCl HS (Sigma-Aldrich, St. Louis, MO) for 30 min, followed by ultrapure water again for 30 min. This cycle was repeated 3 times. Afterwards, corneas were agitated with 0.2% Triton X-100 (TX; Sigma, St. Louis, MO) for 6 hours [20]; 4) NG (Nitrogen Gas) Method: corneas were placed in a 50 ml Falcon tube. Liquid nitrogen was poured into the tubes, closed tightly and parafilmed. NG filled, hypoxic, tubes were kept at room temperature for one week [21]. All corneas were washed with transport solution for 10 minutes, 3 times after decellularization.

Additionally, in order to show the capacity of glycerol to restore transparency in the different methods, three decellularized corneas from each study group were agitated with glycerol (RPI, Mt. Prospect, IL) for two hours.

## 2.2. Characterization

**2.2.1. Macroscopic assessments.** Transparency of decellularized corneas was assessed visually, and the severity of edema was measured using tissue sections. Because of the heterogeneity of the edema, tissue thickness was measured at the shortest and longest places of each cornea. Median sections were used in every assessment.

**2.2.2. Histological analysis.** Hematoxylin-eosin (H&E) staining was performed to evaluate histo-architectural properties of the decellularized corneas. Briefly, corneal samples were fixed in 10% (v/v) neutral buffered formalin for 24h at room temperature, dehydrated, and embedded in paraffin wax. Sections (5μm) were cut and stained with H&E. Afterwards, the sections were analyzed using a light microscope (Leica DMI1, Wetzlar, Germany).

**2.2.3. Nucleic acid analysis.** Hoechst 33342 staining was performed on 5μm tissue sections to evaluate residual nuclear materials. The sections were then observed using a florescence microscope (Leica DMI8, Wetzlar Germany). The number of stained particles was counted by image analysis software (ImageJ, Bethesda, MD) in three identical-sized (0.25 mm$^2$) areas of each corneal stroma.

**2.2.4. Ultrastructural analysis.** Transmission electron microscopy (TEM) imaging was performed to evaluate ultrastructural properties of decellularized corneas. Corneas were fixed with 3% glutaraldehyde in 0.1M phosphate buffer and 80-90nm sections were cut and then viewed on a Tecnai Spirit (Thermo Fisher, Hillsboro, OR) with digital images taken via a charge-coupled device camera (Advanced Microscopy Techniques, Danvers, MA). The average collagen fibril diameters and spacing of the selected electron micrographs were also measured by ImageJ.

**2.2.5. Expression of results and statistical analysis.** All data were expressed as the mean ± SD. Independent samples t test and one-way ANOVA were used to compare mean values. A p value of $<0.05$ was considered as statistically significant. All analyses were performed with commercial software (GraphPad Prism8, San Diego, CA, USA).

## 3. Results

### 3.1. Macroscopic assessments

Although all 3 methods caused edema, SDS caused significantly more edema compared to HS and NG, especially when the limbus was preserved. However, HS caused slightly more edema in corneas without the limbus (Fig 2). The average thickness of the corneas after decellularization increased in all methods in both groups, varying between 20% to 273% (Table 1) compared to untreated corneas, which was 1.15 ± 0.04 mm. NG w/ limbus and SDS w/ limbus groups were able to return to baseline thickness after 2 hours of glycerol treatment, while the other groups were still significantly thicker than the native cornea. Visually, all the methods considerably reduced the transparency. All the groups regained transparency with 2 hours of glycerol treatment, except the SDS w/o limbus group (Fig 2). Since epithelial removal is important for re-epithelialization after transplantation [22], we have further evaluated all corneas with or without limbus in all three decellularization methods and found that epithelial removal was successful in all groups (Fig 3).

### 3.2. Histological examination

WT-NPCs contain a typical lamellar structure as seen in the human cornea, in which the epithelium, Bowman's layer, stroma, Descemet's membrane, and endothelium are all displayed [11]. When treated with SDS, histological examination showed that remarkable numbers of cells were removed inform the corneas, as demonstrated in H&E and Hoechst staining. However,

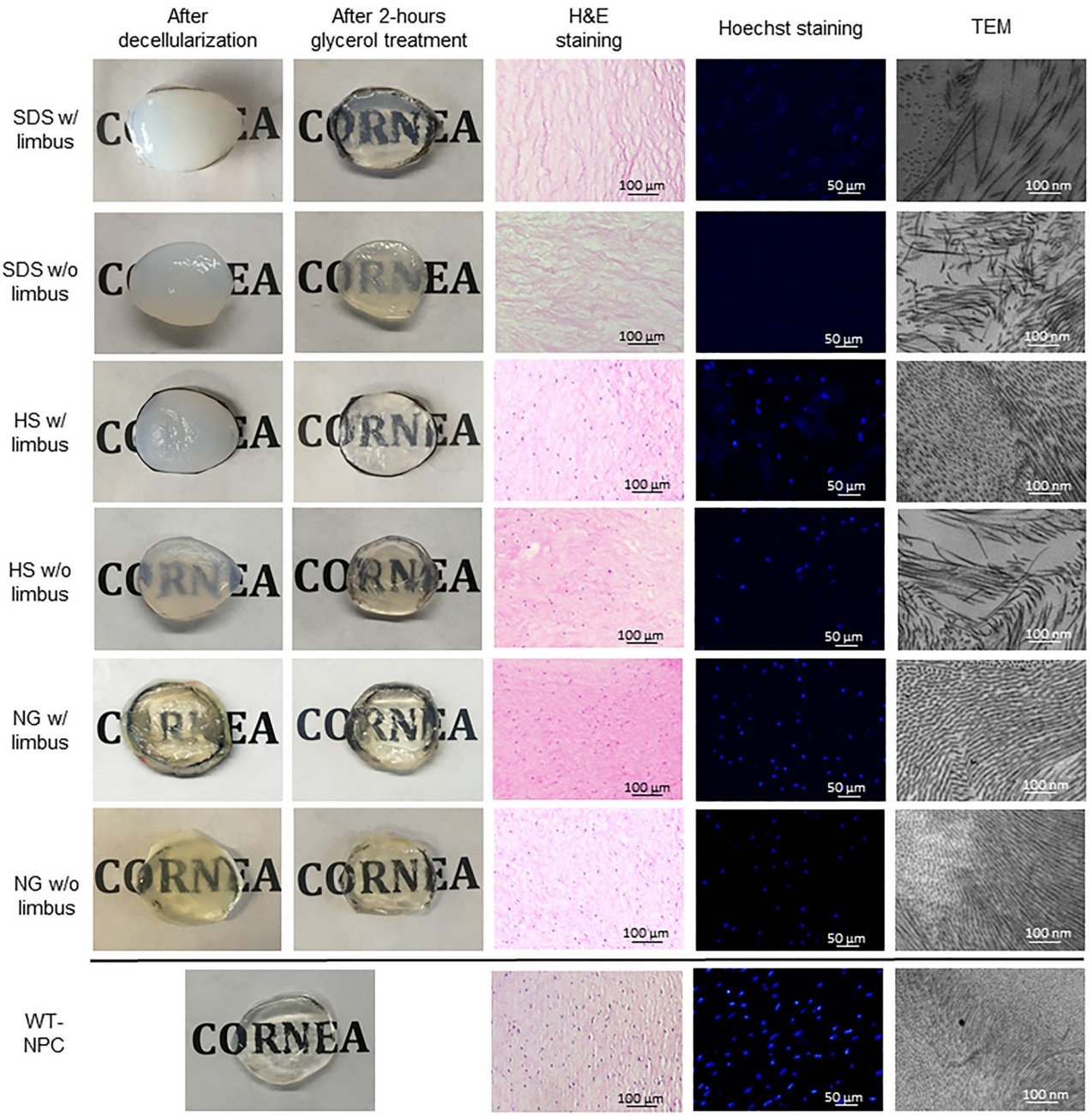

**Fig 2. Macroscopic and microscopic assessments of decellularized and glycerol- treated pig corneas with or without limbus.** HS = Hypertonic saline; H&E = Hematoxylin and eosin; NG = Nitrogen gas; SDS = Sodium dodecyl sulfate; TEM = Transmission electron microscopy; WT-NPC = Wild-type native porcine cornea; w/ = with w/o = without.

many keratocytes were present in HS and NG groups (Fig 2). In the NG group, besides the existence of a large number of cells, even the layers of the cornea were distinguishable.

### 3.3. Nucleic acid analysis

Results of Hoechst staining were mostly consistent with the results of H&E staining. All decellularization methods reduced nuclear components compared to controls (p<0.0001). Fig 2

**Table 1. Average thickness of wild-type porcine corneas after decellularization, after glycerol treatment, and after preservation.**

| | After decellularization | After 2 hours of glycerol treatment | |
| --- | --- | --- | --- |
| | p (Decellularized vs WT-NPC) | p (Decellularized vs glycerol treated) | p (Glycerol treated vs WT-NPC) |
| SDS w/ limbus | 3.62 ± 0.18 | 1.05 ± 0.09 | |
| | <0.0001 | <0.0001 | 0.0524 |
| HS w/ limbus | 1.83 ± 0.14 | 1.44 ± 0.10 | |
| | 0.0009 | 0.0007 | 0.0002 |
| NG w/ limbus | 1.33 ± 0.06 | 1.16 ± 0.06 | |
| | 0.0039 | 0.0018 | 0.7287 |
| SDS w/o limbus | 4.15 ± 0.22 | 1.92 ± 0.22 | |
| | <0.0001 | <0.0001 | <0.0001 |
| HS w/o limbus | 4.29 ± 0.31 | 1.76 ± 0.14 | |
| | <0.0001 | <0.0001 | <0.0001 |
| NG w/o limbus | 1.64 ± 0.11 | 1.33 ± 0.06 | |
| | 0.0033 | 0.0004 | 0.0003 |
| WT—NPC | 1.15 ± 0.04 | | |

All results were expressed as the mean ± SD (millimeter). HS = Hypertonic saline; NG = Nitrogen gas; SDS = Sodium dodecyl sulfate; SD = Standard deviation; WT-NPC = Wild-type native porcine cornea; w/ = with; w/o = without.

shows a representative image from each group. The average of counted nucleic acid particles in a 0.25 mm$^2$ area of the corneal stroma is shown in Table 2. SDS was the strongest agent on the removal of nucleic acids, eliminating approximately 94.3% of nucleic acid in corneas w/ limbus and %100 of nucleic acid in corneas w/o limbus. Although there was a distinct decrease in the amount of nucleic acid detected in HS and NG, a significant amount remained in both groups (Table 2).

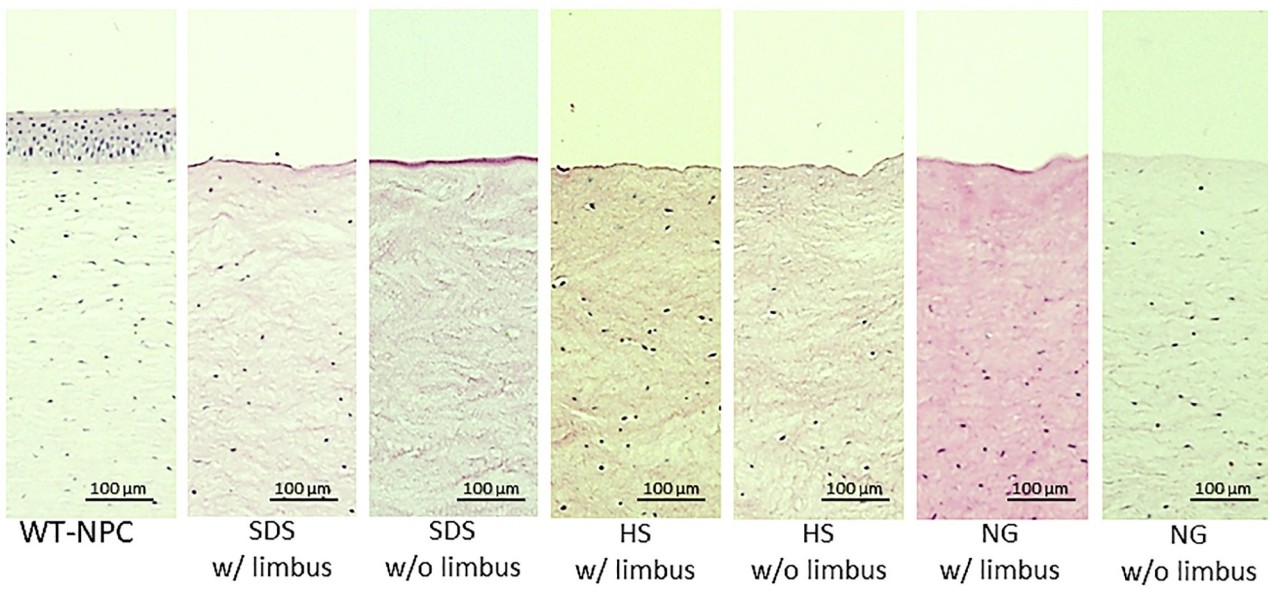

**Fig 3. H&E sections of outermost layers of treated and untreated corneas.** HS = Hypertonic saline; H&E = Hematoxylin and eosin; NG = Nitrogen gas; SDS = Sodium dodecyl sulfate; WT-NPC = Wild-type native porcine cornea; w/ = with w/o = without.

**Table 2. Average number of counted nucleic acid particles in different groups and comparisons.**

| | With Limbus | p (with limbus vs WT—NPC) | Without Limbus | p (without limbus vs WT—NPC) |
|---|---|---|---|---|
| WT-NPC | 162.50 ± 6.68 | | | |
| SDS | 9.25 ± 2.10 | (p<0.0001) | 0 | (p<0.0001) |
| HS | 32.25 ± 3.38 | (p<0.0001) | 41.25 ± 4.05 | (p<0.0001) |
| NG | 87.50 ± 7.24 | (p<0.0001) | 86.25 ± 3.66 | (p<0.0001) |

All results were expressed as the mean ± SD. HS = Hypertonic saline; NG = Nitrogen gas; SD = Standard deviation; SDS = Sodium dodecyl sulfate; WT-NPC = Wild-type native porcine cornea; w/ = with; w/o = without.

## 3.4. Ultrastructural examination

While WT-NPCs had an extremely ordered, stacked collagen fibrillary arrangement, all of the decellularized corneas were changed (Fig 2). Collagen fibril diameters were shown to have significant change between the different groups (Table 3). However, a loss of collagen fibrils, a parallel loss of lamellar structure, and an increase of collagen fibril spacing were observed in all groups (Table 3). SDS disrupted the ultrastructure of the cornea most severely, resulting in disorganization of fibrils, and therefore increased spacing between the fibrils in corneas both with and without limbus.

# 4. Discussion

The conservation of the ECM proteins of the cornea across species favors xenotransplantation over synthetic materials. This advantage makes natural material more suitable for cell attachment, migration, and proliferation. Furthermore, natural material already has the desired shape, transparency and mechanical properties. However, as in other xenotransplantation studies (e.g. liver, heart, kidney), the expression of xenoantigens remains one of the most challenging problems, having the potential to initiate inflammation, rejection, and tissue destruction [23]. Hypothetically, removal of cellular properties from the tissue would reduce the load of antigenicity. In our early studies, we used decellularized full-thickness WT pig corneas and perfused them continuously in a special bioreactor with human serum and peripheral blood mononuclear cells for one week to assess the human immune response to pig corneas which were decellularized by different methods [24]. Although the study was limited, we demonstrated that the HS method resulted in less antibody binding and more biocompatibility, indicating the importance of reducing antigenicity [24].

For the purpose of reducing cellular components and therefore antigenicity, many different decellularization techniques have been utilized to obtain an acellular corneal matrix [11, 21,

**Table 3. Results of the ultrastructural analysis.**

| | Collagen fibril diameter | | Collagen fibril spacing | |
|---|---|---|---|---|
| | w/ limbus, p (w/ limbus vs WT—NPC) | w/o limbus, p (w/o limbus vs WT—NPC) | w/ limbus, p (w/ limbus vs WT—NPC) | w/o limbus, p (w/o limbus vs WT—NPC) |
| SDS | 32.30 ± 1.09 (p = 0.5840) | 31.38 ± 2.77 (p = 0.9678) | 132.64 ± 10.61 (p<0.0001) | 169.09 ± 13.47 (p<0.0001) |
| HS | 31.25 ± 1.04 (p = 0.9073) | 30.98 ± 2.78 (p = 0.7467) | 69.07 ± 5.81 (p<0.0001) | 121.31 ± 6.75 (p<0.0001) |
| NG | 31.63 ± 3.15 (p = 0.8957) | 31.60 ± 3.03 (p = 0.9082) | 38.33 ± 5.82 (p = 0.0343) | 43.62 ± 3.79 (p = 0.0002) |
| WT-NPC | 31.43 ± 3.12 | | 31.27 ± 7.19 | |

All results were expressed as the mean ± SD (nanometer). HS = Hypertonic saline; NG = Nitrogen gas; SD = Standard deviation; SDS = Sodium dodecyl sulfate; WT-NPC = Wild-type native porcine cornea; w/ = with; w/o = without.

25]. The eventual goal of decellularization techniques is to completely remove all cellular components and nuclear materials. Crapo et al. have suggested that it is impossible to remove 100% of cellular material by decellularization techniques. Thus, Crapo et al. have defined the term of "sufficient decellularization" by suggesting minimal criteria, including: (i) ECM should contain <50ng dsDNA per mg dry weight, (ii) DNA fragments should be shorter than 200 bp, and (iii) there should be no visible nuclear component within the ECM [26].

Although these conditions only pertain to cellular and nuclear components, the antigenicity of the cornea is also important. The expression of xenoantigens (e.g., galactose-α1,3-galactose, N-glycolylneuraminic acid) in the cornea is well-known [27]. Lee et al. have previously shown that the reduced antigenicity in decellularized corneas might prolong the survival of the graft [28].

Since the decellularization process removes the endothelium, the efficacy of decellularized grafts is limited to lamellar keratoplasty, unless it is re-endothelialized with human corneal endothelial cells, which currently is challenging [29]. Zhang et al. performed lamellar keratoplasty in 37 patients with fungal keratitis using porcine corneal grafts that were decellularized by the HS method [20]. Neither rejection nor complication was detected in 3-years post-decellularization follow-up. Recently, Shi et al. reported the results of a clinical study indicating the safety of acellular porcine corneas in lamellar keratoplasty for bacterial, fungal, and unknown keratitis [30]. The group used a high hydrostatic pressure-based method to accomplish the decellularization. In the 24-month follow-up, 22 out of 23 grafts survived. Both studies showed promising applications of using acellular porcine corneas in clinical applications, especially in patients who have healthy endothelium and posterior stroma structures. On the other hand, Choi et al. defined the protocol to perform the first clinical trial of penetrating keratoplasty in two patients, by using native porcine corneas with a determined immunosuppressant regimen [31].

In the current study, we used three conventional decellularization methods, demonstrating undesired effects and efficiency of each method, with special attention to the effect of retaining the corneoscleral limbus. SDS, HS, and NG methods were selected for this study because of successful previous reports [19–21]. Among these methods, HS is the only technique which has been employed in a clinical trial [21]. Our results indicated that, although HS had less undesired effects on corneal structure, its decellularization capacity was much lower than SDS. Luo et al. has obtained acellular and ultrastructurally proper corneas using the HS method [32]. Similarly, our results were not consistent with the previous report on the NG method, which pointed out that NG was a promising agent with an ability to decellularize porcine cornea entirely [21]. Many internal (presence of corneoscleral limbus, species of the pig) and external (temperature, ratio of cornea to solution, duration time, concentration of agent, etc.) factors could affect the efficiency of decellularization.

After obtaining a decellularized porcine corneal matrix, the ultimate goal is to recellularize it with human cells. Several different cell types (adipose stem cells, induced pluripotent stem cells, keratocytes, endothelial cells, embryonic stem cells) and various recellularization techniques (seeding, injection) have been used for repopulation of human corneal cells [33–36]. Previously, SDS, HS, and NG based methods created a decellularized cornea that is suitable for recellularization [21, 33, 36]. Future studies will focus on the effect of preserving limbus on recellularization.

## 4.1. Importance of corneoscleral limbus

The corneoscleral limbus is an immunologic and physical barrier between the sclera and cornea [37]. Limbal palisades of Vogt also help to support the unique and richly-vascularized

**Table 4. Comparison of mean values of decellularized corneas with limbus vs without limbus.**

|  | SDS | HS | NG |
|---|---|---|---|
| Thickness after decellularization (from Table 1) | p = 0.0022 | p<0.0001 | p = 0.0004 |
| Thickness after 2 hours glycerol treatment (from Table 1) | p<0.0001 | p = 0.0024 | p = 0.0086 |
| Amount of nucleic acid (from Table 2) | *p<0.0001* | *p = 0.0005* | p = 0.6900 |
| Collagen fibril diameter (from Table 3) | p = 0.5402 | p = 0.8486 | p = 0.9855 |
| Collagen fibril spacing (from Table 3) | p<0.0001 | p<0.0001 | p = 0.0349 |

HS = Hypertonic saline; NG = Nitrogen gas; SD = Standard deviation; SDS = Sodium dodecyl sulfate. Results that written bold are in favor of with limbus group, italic are in favor of without limbus group. More decellularization and better preservation of corneal structure considered as favorable. (In this table each p value represents the comparison of the mean values of decellularized corneas with limbus vs without limbus in respected parameter and method. For example, in the first box, p = 0.0022 represents the comparison of thickness after decellularization between SDS with limbus and SDS without limbus group).

structure of the corneoscleral limbus [38]. The differences between decellularization methods of pig corneas with or without limbus have been summarized in Table 4. According to our results, ultrastructural parameters including thickness after decellularization, thickness after glycerol treatment, and collagen fibril spacing were significantly different between the groups with and without limbus, favoring the group with limbus. However, another structural parameter, collagen fibril diameter, did not show statistically significant change despite the preservation of limbus (Table 3). The decellularization parameter, amount of nucleic acid, was significantly different in the group without limbus in SDS- and HS-based methods but not in NG-based method. Our results indicate that preserving the corneoscleral limbus partially prevented structural damage and edema but would reduce the decellularization capacity (except NG). These differences may originate from the unique structure of the corneoscleral limbus that partly prevents decellularization agents to permeate into the tissue. Ultrastructural and histo-architectural properties are very important to the reversibility of natural corneal properties. However, the extent of decellularization is important to reduce antigenicity as well. Whether preserving the corneoscleral limbus presents a dilemma between greater decellularization and better ultrastructure needs to be further investigated.

## 5. Conclusion

Our findings demonstrated that, among the decellularization methods that were employed, the SDS method in pig corneas without limbus was preferred for its ability to achieve approximately 100% decellularization. Unfortunately, this agent harmed the cornea irreversibly, rendering its clinical use questionable. Although the NG-based method was not associated with important damage to the corneal ultrastructure, its low decellularization capacity (<50%) made it unfavorable. The HS method had a more balanced damage-decellularization ratio, but its decellularization capacity was lower than the SDS method. Future clinical trials should investigate (i) the human immune response against decellularized porcine corneas in a more detailed way, and (ii) the long-term physical and biological properties of xeno-transplanted porcine corneas.

## Supporting information

**S1 Data.**
(XLSX)

**S2 Data.**
(XLSX)

**S3 Data.**
(XLSX)

**S4 Data.**
(XLSX)

**S5 Data.**
(XLSX)

## Author Contributions

**Conceptualization:** Abdulkadir Isidan, Shaohui Liu, Hidetaka Hara.

**Data curation:** Abdulkadir Isidan, Shaohui Liu.

**Formal analysis:** Abdulkadir Isidan.

**Funding acquisition:** Burcin Ekser.

**Investigation:** Abdulkadir Isidan, Shaohui Liu, Angela M. Chen, Wenjun Zhang, Ping Li, Burcin Ekser.

**Methodology:** Abdulkadir Isidan, Angela M. Chen, Wenjun Zhang, Ping Li, Lester J. Smith, Burcin Ekser.

**Project administration:** Abdulkadir Isidan.

**Resources:** Wenjun Zhang, Burcin Ekser.

**Software:** Abdulkadir Isidan, Lester J. Smith.

**Supervision:** Abdulkadir Isidan, Shaohui Liu, Wenjun Zhang, Ping Li, Lester J. Smith, Hidetaka Hara, David K. C. Cooper.

**Validation:** Abdulkadir Isidan, Shaohui Liu, Wenjun Zhang, Ping Li, David K. C. Cooper.

**Visualization:** Abdulkadir Isidan, Shaohui Liu, Wenjun Zhang, Ping Li.

**Writing – original draft:** Abdulkadir Isidan, Shaohui Liu, Angela M. Chen, Hidetaka Hara, Burcin Ekser.

**Writing – review & editing:** Abdulkadir Isidan, Shaohui Liu, Angela M. Chen, Wenjun Zhang, Ping Li, Lester J. Smith, Hidetaka Hara, David K. C. Cooper, Burcin Ekser.

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
