## [Decision Letter · Decision Letter 0]

29 Dec 2020

PONE-D-20-37048

Comparison of porcine corneal decellularization methods and importance of preserving corneal limbus through decellularization

PLOS ONE

Dear Dr. Ekser,

Thank you for submitting your manuscript to PLOS ONE. After careful consideration, we feel that it has merit but does not fully meet PLOS ONE’s publication criteria as it currently stands. Therefore, we invite you to submit a revised version of the manuscript that addresses the points raised during the review process.

The paper has been reviewed and a number of concerns voiced. Please revise the paper and pay particular attention to the following points:

1. Please elaborate on the finding that preserving the limbus prevented structural damage and edema, but reduced the decellularization efficiency.

2. Please discuss/perform recellularization as a functional test for decellularization method that would preserve tissue architecture and ECM.

3. Please present data on evaluating epithelial removal, which is important for transplantation of recellularized cornea. The choice of decellularization method may be critical: Shafiq MA et al. 2012, Decellularized Human Cornea for Reconstructing the Corneal Epithelium and Anterior Stroma. Tissue Engineering Part C Methods 18:340.

4. Please revisit section 3.4 for discrepancy with Table 3 in relation for collagen fibril diameter changes for all decellularization methods.

5. Please better explain the data in Table 4 as suggested.

6. Please add and discuss recent papers mentioned by the reviewer and also include data on human corneas for comparison.

We look forward to receiving your revised manuscript.

Kind regards,

Alexander V. Ljubimov, Ph.D.

Academic Editor

PLOS ONE

Journal Requirements:

"Work on xenotransplantation in the Xenotransplantation Research Laboratory at Indiana

University has been supported by internal funds of the Department of Surgery, in part,

with support by the Board of Directors of the Indiana University Health Values Fund for

Research Award (VFR-457-Ekser), the Indiana Clinical and Translational Sciences

Institute, funded in part by Grant # UL1TR001108 from the National Institutes of Health,

National Center for Advancing Translational Sciences, Clinical and Translational

Sciences Award, and by the special research agreement with Lung Biotechnology LLC

and United Therapeutics Corp, Silver Spring, MD, USA."

Additionally, because some of your funding information pertains to commercial funding, we ask you to provide an updated Competing Interests statement, declaring all sources of commercial funding.

In your Competing Interests statement, please confirm that your commercial funding does not alter your adherence to PLOS ONE Editorial policies and criteria by including the following statement: "This does not alter our adherence to PLOS ONE policies on sharing data and materials.” as detailed online in our guide for authors  http://journals.plos.org/plosone/s/competing-interests.  If this statement is not true and your adherence to PLOS policies on sharing data and materials is altered, please explain how.

Please include the updated Competing Interests Statement and Funding Statement in your cover letter. We will change the online submission form on your behalf.

Reviewers' comments:

Reviewer's Responses to Questions

**Comments to the Author**

1. Is the manuscript technically sound, and do the data support the conclusions?

Reviewer #1: Partly

2. Has the statistical analysis been performed appropriately and rigorously? 

Reviewer #1: Yes

3. Have the authors made all data underlying the findings in their manuscript fully available?

Reviewer #1: Yes

4. Is the manuscript presented in an intelligible fashion and written in standard English?

Reviewer #1: Yes

5. Review Comments to the Author

Reviewer #1: The study presented appears to be an extension of the previous study that examined the same three methods of porcine cornea decellularization, along with freezing-thawing method, and that has not been referred to in this manuscript (https://cm.ixa2019.org/webapp/lecture/233). In both studies the conclusion was similar (whereas the specific data were somewhat different between the two), i.e. none of the methods tested can be considered effective and optimal. The further development of the present study is the focus on a possible role of the limbal region for the efficacy of cell removal and better preservation of the cornea.

There are some comments to consider:

1. It was found that preserving the limbus partially prevented structural damage and edema, but also reduced the decellularization efficiency. This interesting effect of the limbus should probably be discussed more than it was on p.15 (lines 1-3).

2. To find a better method of decellularization, it may be worthy to use some sort of functional test, such as evaluation of re-cellularization of decellularized porcine cornea that may depend on preservation of the tissue architecture and ECM and thus would provide ultimate criterion for developing xenotransplant.

3. No data were presented on evaluating epithelial removal which may be important for transplantation of re-cellularized cornea. From the standpoint of successful corneal cell repopulation, the choice of the cornea decellularization may be critical as shown before for human corneas (Shafiq MA et al. 2012, Decellularized Human Cornea for Reconstructing the Corneal Epithelium and Anterior Stroma. Tissue Engineering Part C Methods 18:340).

4. In Section 3.4, it appears that the data description (p.11, line 7) is incorrect, as collagen fibril diameters showed no significant changes (Table 3) for all decellularization methods.

Also, the increase of collagen fibril spacing were observed not in all groups as stated (p.11, line 9), because in “NG with limbus” group the change is not that significant (p=0.034, Table 3).

5. It is not clear how mean values in Table 4 were obtained and what did “the results in bold or italic are in favor of with limbus/ or without limbus” (p.16, l.1-3) exactly mean, so please, elaborate.

6. The list of cited literature (21 ref. long) is not full enough, and some recent papers are missing. It is probably worthwhile to also include human cornea data which are rather numerous. More specifically, promising data were reported on the use of Benzonase endonuclease alone (Liu J et al. Application of benzonase in preparation of decellularized lamellar porcine corneal stroma for lamellar keratoplasty. J. Biomed. Mater. Res. Part A 2019, 107, 2547) or Benzonase combined with SDS treatment (Alio del Barrio JL et al. 2015. Acellular human corneal matrix sheets seeded with human adipose-derived mesenchymal stem cells integrate functionally in an experimental animal model. Exp. Eye Res. 132, 91) provided minimal loss of optical transparency and good results in animal transplantation assays. Similar combined decellularization approach using sodium N-lauroyl surfactant and supernuclease resulted in good transparency and biocompatibility without degradation 4 weeks after transplantation (Dong, M. et al., Rapid porcine corneal decellularization through the use of sodium N-lauroyl glutamate and supernuclease. J. Tissue Eng. 2019, 10). Also, a recent review contained relevant corneal data (Mendibil U et al., Tissue-Specific Decellularization Methods: Rationale and Strategies to Achieve Regenerative Compounds. Int. J. Mol. Sci. 2020, 21, 5447; doi:10.3390/ijms21155447).

6. PLOS authors have the option to publish the peer review history of their article (what does this mean?). If published, this will include your full peer review and any attached files.

Reviewer #1: No

---

## [Author Response · Author response to Decision Letter 0]

22 Jan 2021

RESPONSE LETTER - PONE-D-20-37048

Comparison of porcine corneal decellularization methods and importance of preserving corneal limbus through decellularization

ACADEMIC EDITOR’S COMMENTS

1. Please elaborate on the finding that preserving the limbus prevented structural damage and edema, but reduced the decellularization efficiency.

- We have now further elaborated on the impact of preserving the limbus which prevented structural damage and edema. Please see below responses to reviewers. 

2. Please discuss/perform recellularization as a functional test for decellularization method that would preserve tissue architecture and ECM.

- The scope of the current manuscript is to compare decellularization methods side by side. Recellularization is not within the scope of the manuscript. However, we have further discussed possible recellularization in the discussion section. (see also below).

3. Please present data on evaluating epithelial removal, which is important for transplantation of recellularized cornea. The choice of decellularization method may be critical: Shafiq MA et al. 2012, Decellularized Human Cornea for Reconstructing the Corneal Epithelium and Anterior Stroma. Tissue Engineering Part C Methods 18:340.

- We have now further evaluated epithelial removal, as suggested. We have also added a new figure. Please see below response to the reviewer.

4. Please revisit section 3.4 for discrepancy with Table 3 in relation for collagen fibril diameter changes for all decellularization methods.

- We have now corrected this discrepancy (please see below).

5. Please better explain the data in Table 4 as suggested.

- Please see below – response to the reviewer.

6. Please add and discuss recent papers mentioned by the reviewer and also include data on human corneas for comparison.

- We have now added all suggested papers. 

Reviewer #1: The study presented appears to be an extension of the previous study that examined the same three methods of porcine cornea decellularization, along with freezing-thawing method, and that has not been referred to in this manuscript (https://cm.ixa2019.org/webapp/lecture/233). 

- We thank the reviewer for her/his comments. The link belongs to OUR initial observation which was presented at the International Xenotransplantation Association Meeting as an abstract. 

In both studies the conclusion was similar (whereas the specific data were somewhat different between the two), i.e. none of the methods tested can be considered effective and optimal. The further development of the present study is the focus on a possible role of the limbal region for the efficacy of cell removal and better preservation of the cornea.

- As noted by the reviewer, the link belongs to an abstract where only OUR initial observations were presented. We have further analyzed and increased the ‘n’ to write the current manuscript. In fact, the reviewer also noted that in the abstract there was no information regarding limbus since that was a research in progress and we have presented our complete data in the current manuscript.

1. It was found that preserving the limbus partially prevented structural damage and edema, but also reduced the decellularization efficiency. This interesting effect of the limbus should probably be discussed more than it was on page 15 (lines 1-3).

- We have now extended the “Importance of corneoscleral limbus” section by further discussing the results of Table 4 in details. We have specifically added/changed below comments in regarded section.

“According to our results, ultrastructural parameters including thickness after decellularization, thickness after glycerol treatment, and collagen fibril spacing were significantly different between the groups with and without limbus, favoring the group with limbus. However, another structural parameter, collagen fibril diameter, did not show statistically significant change despite the preservation of limbus (Table 3). The decellularization parameter, amount of nucleic acid, was significantly different in the group without limbus in SDS- and HS-based methods but not in NG-based method. Our results indicate that preserving the corneoscleral limbus partially prevented structural damage and edema but would reduce the decellularization capacity (except NG). These differences may originate from the unique structure of the corneoscleral limbus that partly prevents decellularization agents to permeate into the tissue.”

2. To find a better method of decellularization, it may be worthy to use some sort of functional test, such as evaluation of re-cellularization of decellularized porcine cornea that may depend on preservation of the tissue architecture and ECM and thus would provide ultimate criterion for developing xenotransplant.

- We agree with the reviewer that re-cellularization of decellularized porcine cornea may be a test to further evaluate the functionality. The scope of the current manuscript is to compare side-by-side 3 decellularization techniques with or without limbus (which was never done before). Evaluating re-epithelialization, stromal re-cellularization and re-endothelialization between with limbus and without limbus groups are of our interests and definitely among our future plans. However, due to our current situation and strict limitation in basic science research due to COVID-19 at academic institutes, we will not be able to perform recellularization experiments for foreseeable future. 

- Because of the importance and up-to-date-ness of the matter, we have now added the below paragraph to the discussion section where we have further discussed recellularization and possible effects of preserving limbus on recellularization with recent publications. 

“After obtaining a decellularized porcine corneal matrix, the ultimate goal is to recellularize it with human cells. Several different cell types (adipose stem cells, induced pluripotent stem cells, keratocytes, endothelial cells, embryonic stem cells) and various recellularization techniques (seeding, injection) have been used for repopulation of human corneal cells. Previously, SDS, HS, and NG based methods created a decellularized cornea that is suitable for recellularization. Future studies will focus on the effect of preserving limbus on recellularization.”

3. No data were presented on evaluating epithelial removal which may be important for transplantation of re-cellularized cornea. From the standpoint of successful corneal cell repopulation, the choice of the cornea decellularization may be critical as shown before for human corneas (Shafiq MA et al. 2012, Decellularized Human Cornea for Reconstructing the Corneal Epithelium and Anterior Stroma. Tissue Engineering Part C Methods 18:340).

- We thank the reviewer for her/his comments. We agree with the reviewer that it is an important point. In order to evaluate the epithelial removal, we have now further analyzed the decellularized epitheliums in all three decellularization methods (SDS, HS, NG) with or without limbus. We have now added a new figure (Figure 3) and new results have been added to the result section on page 8.

4. In Section 3.4, it appears that the data description (p.11, line 7) is incorrect, as collagen fibril diameters showed no significant changes (Table 3) for all decellularization methods.

- We thank the reviewer for this observation. We have now omitted the part of “resulting in a loss of fibrils” and then re-wrote the sentence as below.

“SDS disrupted the ultrastructure of the cornea most severely, resulting in disorganization of fibrils, and therefore increased spacing between the fibrils in corneas both with and without limbus.”

Also, the increase of collagen fibril spacing were observed not in all groups as stated (p.11, line 9), because in “NG with limbus” group the change is not that significant (p=0.034, Table 3).

- We have considered p value <0.05 is a statistically significant finding. Therefore, abovementioned p value (p=0.034) was a statistically significant finding for us. We apologize but we did not understand the reviewer’s comment on “not that significant”. 

5. It is not clear how mean values in Table 4 were obtained and what did “the results in bold or italic are in favor of with limbus/ or without limbus” (p.16, l.1-3) exactly mean, so please, elaborate.

- We thank to the reviewer for pointing out the ambiguity in Table 4. We have now improved and elaborated the meaning of Table 4 by adding further explanation in the legend. We have also mentioned where we obtained the mean values in the first column of the table.

“More decellularization and better preservation of corneal structure considered as favorable. (In this table each p value represents the comparison of the mean values of decellularized corneas with limbus vs without limbus in respected parameter and method. For example, in the first box, p=0.0022 represents the comparison of thickness after decellularization between SDS with limbus and SDS without limbus group.)”

6. The list of cited literature (21 ref. long) is not full enough, and some recent papers are missing. It is probably worthwhile to also include human cornea data which are rather numerous. More specifically, promising data were reported on the use of Benzonase endonuclease alone (Liu J et al. Application of benzonase in preparation of decellularized lamellar porcine corneal stroma for lamellar keratoplasty. J. Biomed. Mater. Res. Part A 2019, 107, 2547) or Benzonase combined with SDS treatment (Alio del Barrio JL et al. 2015. Acellular human corneal matrix sheets seeded with human adipose-derived mesenchymal stem cells integrate functionally in an experimental animal model. Exp. Eye Res. 132, 91) provided minimal loss of optical transparency and good results in animal transplantation assays. Similar combined decellularization approach using sodium N-lauroyl surfactant and supernuclease resulted in good transparency and biocompatibility without degradation 4 weeks after transplantation (Dong, M. et al., Rapid porcine corneal decellularization through the use of sodium N-lauroyl glutamate and supernuclease. J. Tissue Eng. 2019, 10). Also, a recent review contained relevant corneal data (Mendibil U et al., Tissue-Specific Decellularization Methods: Rationale and Strategies to Achieve Regenerative Compounds. Int. J. Mol. Sci. 2020, 21, 5447; doi:10.3390/ijms21155447).

- We thank to the reviewer for suggesting up-to-date papers for our article. We have now added all suggested papers as well as couple more from 2019 and 2020. Below please find the new references.

12 - Mendibil U, Ruiz-Hernandez R, Retegi-Carrion S, Garcia-Urquia N, Olalde-Graells B, Abarrategi A. Tissue-Specific Decellularization Methods: Rationale and Strategies to Achieve Regenerative Compounds. Int J Mol Sci. 2020;21(15):5447.

13 - Sasaki S, Funamoto S, Hashimoto Y, Kimura T, Honda T, Hattori S, et al. In vivo evaluation of a novel scaffold for artificial corneas prepared by using ultrahigh hydrostatic pressure to decellularize porcine corneas. Mol Vis. 2009;15:2022-2028.

14 - Topuz B, Günal G, Guler S, Aydin HM. Use of supercritical CO2 in soft tissue decellularization. Methods Cell Biol. 2020;157:49-79.

15 - Wu Z, Zhou Y, Li N, Huang M, Duan H, Ge J, et al. The use of phospholipase A(2) to prepare acellular porcine corneal stroma as a tissue engineering scaffold. Biomaterials. 2009;30(21):3513-22.

16 - Lin HJ, Wang TJ, Li TW, Chang YY, Sheu MT, Huang YY, et al. Development of Decellularized Cornea by Organic Acid Treatment for Corneal Regeneration. Tissue Eng Part A. 2019;25(7-8):652-662.

17 - Liu J, Li Z, Li J, Liu Z. Application of benzonase in preparation of decellularized lamellar porcine corneal stroma for lamellar keratoplasty. J Biomed Mater Res A. 2019;107(11):2547-2555.

18 - Dong M, Zhao L, Wang F, Hu X, Li H, Liu T, et al. Rapid porcine corneal decellularization through the use of sodium N-lauroyl glutamate and supernuclease. J Tissue Eng. 2019;10:2041731419875876.

22 - Shafiq MA, Gemeinhart RA, Yue BY, Djalilian AR. Decellularized human cornea for reconstructing the corneal epithelium and anterior stroma. Tissue Eng Part C Methods. 2012;18(5):340-8.

33 - Alio del Barrio JL, Chiesa M, Garagorri N, Garcia-Urquia N, Fernandez-Delgado J, Bataille L, et al. Acellular human corneal matrix sheets seeded with human adipose-derived mesenchymal stem cells integrate functionally in an experimental animal model. Exp Eye Res. 2015;132:91-100.

34 - Fernández-Pérez J, Ahearne M. Decellularization and recellularization of cornea: Progress towards a donor alternative. Methods. 2020;171:86-96.

35 - Alió Del Barrio JL, El Zarif M, Azaar A, Makdissy N, Khalil C, Harb W, et al. Corneal Stroma Enhancement With Decellularized Stromal Laminas With or Without Stem Cell Recellularization for Advanced Keratoconus. Am J Ophthalmol. 2018;186:47-58.

36 - Da Mata Martins TM, da Silva Cunha P, Rodrigues MA, de Carvalho JL, de Souza JE, de Carvalho Oliveira JA, et al. Epithelial basement membrane of human decellularized cornea as a suitable substrate for differentiation of embryonic stem cells into corneal epithelial-like cells. Mater Sci Eng C. 2020;116:111215.

We thank the editors and reviewers for their positive comments, questions, and feedbacks. We believe that the revised version is now much better than the original submission with their inputs. We hope that the revised version is now acceptable for publication.

Yours sincerely,

Abdulkadir Isidan, MD

Burcin Ekser, MD, PhD

(on behalf of all coauthors).

---

## [Decision Letter · Decision Letter 1]

3 Feb 2021

Comparison of porcine corneal decellularization methods and importance of preserving corneal limbus through decellularization

PONE-D-20-37048R1

Dear Dr. Ekser,

We’re pleased to inform you that your manuscript has been judged scientifically suitable for publication and will be formally accepted for publication once it meets all outstanding technical requirements.

Kind regards,

Alexander V. Ljubimov, Ph.D.

Academic Editor

PLOS ONE

Additional Editor Comments (optional):

Reviewers' comments:

Reviewer's Responses to Questions

**Comments to the Author**

1. If the authors have adequately addressed your comments raised in a previous round of review and you feel that this manuscript is now acceptable for publication, you may indicate that here to bypass the “Comments to the Author” section, enter your conflict of interest statement in the “Confidential to Editor” section, and submit your "Accept" recommendation.

Reviewer #1: All comments have been addressed

2. Is the manuscript technically sound, and do the data support the conclusions?

Reviewer #1: (No Response)

3. Has the statistical analysis been performed appropriately and rigorously? 

Reviewer #1: (No Response)

4. Have the authors made all data underlying the findings in their manuscript fully available?

Reviewer #1: (No Response)

5. Is the manuscript presented in an intelligible fashion and written in standard English?

Reviewer #1: (No Response)

6. Review Comments to the Author

Reviewer #1: (No Response)

7. PLOS authors have the option to publish the peer review history of their article (what does this mean?). If published, this will include your full peer review and any attached files.

Reviewer #1: No

---

## [Editor Report · Acceptance letter]

23 Feb 2021

PONE-D-20-37048R1 

Comparison of porcine corneal decellularization methods and importance of preserving corneal limbus through decellularization 

Dear Dr. Ekser:

I'm pleased to inform you that your manuscript has been deemed suitable for publication in PLOS ONE. Congratulations! Your manuscript is now with our production department. 

Kind regards, 

on behalf of

Dr. Alexander V. Ljubimov 

Academic Editor

PLOS ONE